# EPOS trial: the effect of air filtration through a plasma chamber on the incidence of surgical site infection in orthopaedic surgery: a study protocol of a randomised, double-blind, placebo-controlled trial

Anders Persson ![ORCID],[1] Isam Atroshi ![ORCID],[2,3] Thomas Tyszkiewicz,[2,3] Nils Hailer,[4,5] Stergios Lazarinis,[4,5] Thomas Eisler,[1] Harald Brismar,[6,7] Sebastian Mukka,[8,9] Per-Juan Kernell,[10] Maziar Mohaddes,[11,12] Olof Sköldenberg,[1] Max Gordon[1]

For numbered affiliations see end of article.

**Correspondence to**
Dr Anders Persson;
anders.persson@
regionstockholm.se

## ABSTRACT

**Introduction** There is controversy regarding the importance of air-transmitted infections for surgical site infections (SSIs) after orthopaedic surgery. Research has been hindered by both the inability in blinding the exposure, and by the need for recruiting large enough cohorts. The aim of this study is to investigate whether using a new form of air purifier using plasma air purification (PAP) in operating rooms (ORs) lowers the SSI rate or not.

**Methods and analysis** Multicentre, double-blind, cluster-randomised, placebo-controlled trial conducted at seven hospitals in 2017–2022. All patients that undergo orthopaedic surgery for minimum 30 min are included. Intervention group: patients operated in OR with PAP devices turned on. Control group: patients operated in OR with PAP devices turned off. Randomisation: each OR will be randomised in periods of 4 weeks, 6 weeks or 8 weeks to either have the devices on or off. Primary outcome: any SSI postoperatively defined as a composite endpoint of any of the following: use of isoxazolylpenicillin, clindamycin or rifampicin for 2 days or more, International Classification of Diseases codes or Nordic Medico-Statistical Committee codes indicating postoperative infection. In a second step, we will perform a chart review on those patients with positive indicators of SSI to further validate the outcome. Secondary outcomes are described in the Methods section. Power: we assume an SSI rate of 2%, an SSI reduction rate of 25% and we need approximately 45 000 patients to attain a power of 80% at a significance level of 0.05.

**Ethics and dissemination** The study is approved by the Swedish Ethical Review Authority. The interim analysis results from the study will be presented only to the researchers involved unless the study thereafter is interrupted for whatever reason. Publication in a medical journal will be presented after inclusion of the last patient.

**Trial registration number** NCT02695368.

## Strengths and limitations of this study

► EPOS is a multicentre, placebo-controlled trial with approximately 45 000 study subjects, that will evaluate the effect of plasma air purification on the incidence of surgical site infections (SSIs) after orthopaedic surgery.
► The double-blinded design provides strong internal validity to the study results.
► The cluster-randomisation design, which is created through switching of the exposure within each operating room (OR), minimises the risk of allocation bias.
► This study is the first randomized controlled trial, to the best of our knowledge, investigating the true cause-and-effect relationship between an air-purifying intervention in ORs and SSIs.
► The primary limitation to the study is the resource intensity, mainly due to the large number of study subjects required to study such an unusual outcome, and the concomitant review of medical records to validate the outcome.

## INTRODUCTION

Despite surgery in clean operating rooms (ORs), surface sterilisation and antibiotics, SSI after orthopaedic surgery have an overall estimated incidence of 1%–4%.[1–3] This feared complication is associated with long-term antibiotics, repeated surgeries, prolonged hospital stays, economic burden and a poorer end result for individual patients.[4 5] Prevention of SSIs is, therefore, of paramount importance.

Air flow within the OR can spread airborne particles, posing a potential risk for postoperative infection. These airborne particles

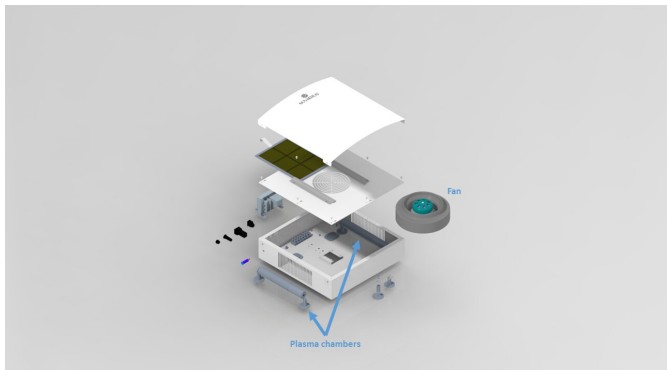

**Figure 1** Rendered view of Novaerus NV800.

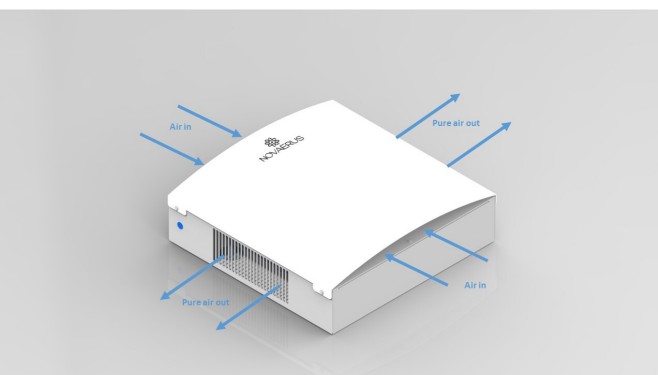

**Figure 2** Air flow through the air purifier.

include dust, textile fibres, skin scales and respiratory aerosols, loaded with viable microorganisms (including *Staphylococcus aureus*) having been released mainly from the surgical team members and patient into the surrounding air of the OR. These particles have been shown to settle onto surfaces, including the surgical wound and instruments.[6] Thus, air-transmitted infections is one of the main reasons for SSI,[6] making this an interesting intervention target.

Ever since the ground-breaking study by Charnley and Eftekhar,[7] which showed that cleaner air in the OR drastically improved infection rates after total hip arthroplasty, vast efforts have been made to address this issue. Lidwell *et al* carried on their work, showing that ultra-clean air (ie, <10 colony-forming units (CFUs)/m$^3$) in combination with body-exhaust suits and prophylactic antibiotics further improved infection rates.[8–10] Based on their work, in combination with several reports showing that laminar air flow (LAF) reduces bacterial contamination in the OR air,[11–13] many ORs are today equipped with LAF ventilation. Unfortunately, though, when evaluated in large cohort studies, systematic reviews and meta-analyses, LAF systems in modern state-of-the-art ORs have so far failed to prove efficient in preventing infections, compared with conventional ventilation.[14–19]

The plasma air purification (PAP) system used in our current study is an air purifier that sterilises the air particles through a plasma chamber (figure 1). Air in the OR is pumped through the chamber, and by using a small current, it transforms the air in the vicinity of the electrode into plasma, which eradicates any bacteria that pass through (figure 2). The small size of the machine allows it to fit into any operating theatre without interfering with existing equipment.[20]

Similar systems exists that use ultraviolet light to sterilise air particles and reduce the rate of hospital-acquired infections.[21 22] There are though few peer-reviewed articles regarding air purification. While we have not found any randomised clinical trials, there are randomised field trials, outside hospital settings, that have shown positive effects in vivo. In a blinded randomised field trial on healthy volunteers using air purifiers, a significant reduction in air particles were seen and this also led to a reduction of stress hormone for the participants.[23]

In hospital and OR settings, the PAP technology alone significantly reduces the number of CFUs of *Staphylococci* (the most common infecting microorganism in SSI) from 49% to 97%.[24] In non-randomised studies, it has been shown to reduce respiratory infections, personnel sick leave and severe infectious outbreaks.[22 24] These effects have though not been validated in randomised clinical trials in OR settings. However, this is also true for all methods of reducing airborne pathogens in ORs that currently are used, such as surgery in LAF ORs. The need for Level 1 evidence in this field of medicine is urgent.

The EPOS trial is, therefore, designed as a multicentre, double-blinded, cluster-randomised, placebo-controlled trial.

The aim of this study is to investigate whether using PAP devices that clean the air from contaminating particles in ORs lower the rate of SSI, or not. The primary endpoint will be SSI rate within 12 weeks postoperatively, defined as either use of antibiotics targeting common implant pathogens, International Classification of Diseases (ICD) code or Nordic Medico-Statistical Committee (NOMESCO) code indicating postoperative infection. This proxy variable is more closely described in the Methods section. We hypothesise that PAP can reduce the incidence of SSI in orthopaedic surgery by 25%. Secondary aims include investigating the number of prescribed antibiotics as well as the number of needed readmissions and length of stay for SSI.

## METHODS: PARTICIPANTS, INTERVENTIONS AND OUTCOMES
### Study setting
The study is being conducted at seven major hospitals in Sweden (table 1). Inclusion will take place between April 2017 and 31 December 2021. The randomised clinical trial setting has been chosen to control for the huge number of possible confounders influencing the outcome. The study protocol has been written according to the Standard Protocol Items: Recommendations for Interventional Trials statement. A Consolidated Standards of Reporting Trials flow diagram, published as an

**Table 1** Recruitment centres and estimated recruitment based on number of surgeries performed annually. The hospitals are already recruiting patients

| Centre | Estimated n | % recruited |
|---|---|---|
| Danderyd Hospital | 8000 | 18% |
| Hässleholm/Kristianstad | 10 000 | 22% |
| Huddinge Hospital | 6000 | 13% |
| Akademiska Hospital | 10 000 | 22% |
| Ortho Centre | 3000 | 7% |
| Umeå | 8000 | 18% |
| Total sample size | 45 000 | 100% |

online supplemental file 1 to this protocol, describes our study graphically.

### Eligibility criteria

We will include[1] all patients that undergo surgery for 30 min or longer at each centre during the study period. We assume that surgeries lasting less than 30 min are less susceptible to SSIs. Including those in this study would, therefore, result in a larger cohort. We will exclude surgeries on (1) already infected surgical sites, defined as: ICD or NOMESCO codes indicating infection (same as those used for the primary outcome, see below), open fractures, traumatic wounds and vacuum-assisted wound therapy, (2) patients that have withdrawn antibiotics 2 weeks or less prior to surgery and (3) patients that have actively marked their hospital charts with an added privacy notice. If patients have multiple surgeries during the study period, only the first operation will be included.

### Intervention

At each hospital, all ORs that perform surgery on orthopaedic patients will be equipped with three[3] PAP systems each. The groups are defined as: intervention group: those operated where the PAP device has been turned on for at least 2 days prior to index surgery; control group: those where the PAP device has been turned off for at least 2 days prior to index surgery; and mixed group: those receiving surgeries in ORs within 2 days after the PAP device switches status. In the analysis we will also subgroup the study subjects according to measurements prior to study start into regular ORs ($\geq 10$ CFU/m$^3$) and ultra-clean ORs (<10 CFU/m$^3$).

PAP device status and function will be monitored continuously during the inclusion period through standardised manual controls every 3 months, and also at the end of the inclusion period by validating PAP device status retrospectively through memory card recordings in each device.

### Outcomes

The primary outcome is any indication of SSI within 12 weeks postoperatively, defined as a composite endpoint of any of the following:

**Box 1 CDC definition of SSI**

**Superficial incisional SSI**
Infection within 30 days after the operation and only involves skin and subcutaneous tissue of the incision and at least one of the following:
► Purulent drainage with or without laboratory confirmation, from the superficial incision.
► Organisms isolated from an aseptically obtained culture of fluid or tissue from the superficial incision.
► At least one of the following signs or symptoms of infection: pain or tenderness, localised swelling, redness or heat, and superficial incision is deliberately opened by surgeon, unless incision is culture-negative.
► Diagnosis of superficial incisional SSI made by a surgeon or attending physician.

**Deep incisional SSI**
Infection occurs within 30 days after the operation if no implant is left in place or within 1 year if implant is in place and the infection appears to be related to the operation and infection involves deep soft tissue (eg, fascia and muscle) of the incision and at least one of the following:
► Purulent drainage from the deep incision but not from the organ/space component of the surgical site.
► A deep incision spontaneously dehisces or is deliberately opened by a surgeon when the patient has at least one of the following signs or symptoms: fever (>38°C), localised pain or tenderness, unless incision is culture-negative.
► An abscess or other evidence of infection involving the deep incision is found on direct examination, during reoperation, or by histopathologic or radiologic examination.
► Diagnosis of deep incisional SSI made by a surgeon or attending physician.

**Organ/space SSI**
Infection occurs within 30 days after the operation if no implant is left in place or within 1 year if implant is in place and the infection appears to be related to the operation and infection involves any part of the anatomy (eg, organs and spaces) other than the incision, which was opened or manipulated during an operation and at least one of the following:
► Purulent drainage from a drain that is placed through a stab wound into the organ/space.
► Organisms isolated from an aseptically obtained culture of fluid or tissue in the organ/space.
► An abscess or other evidence of infection involving the organ/space that is found on direct examination, during reoperation, or by histopathologic or radiologic examination.
► Diagnosis of organ/space SSI made by a surgeon or attending physician.

SSI, surgical site infection.

1. Withdrawal or other documented use of antibiotics corresponding to 2 days or more after surgery targeting *S. aureus*.
2. ICD code indicating postoperative infection (at date of readmission)
3. NOMESCO code indicating postoperative infection.

In a second step, we will perform a chart review on those patients with positive indicators of SSI to further validate the outcome. We will use the internationally accepted CDC definition of SSI when finally establishing that an SSI has occurred (box 1).[25]

The effect size of the primary endpoint is the relative risk of contracting SSI for the intervention group versus the control group and will be calculated by dividing the probability of contracting an SSI in the intervention group by the control group=$\frac{SSI\ rate\ intervention\ group}{SSI\ rate\ control\ group}$. The effect size will also be presented as an absolute risk difference=$SSI\ rate\ intervention\ group - SSI\ rate\ control\ group$.

The primary outcome is a surrogate variable for SSI as it, due to the large sample size, is practically impossible to have all patients come back for outpatient visits and be visually inspected. To further validate the outcome, a medical record review will be performed in a second step on all individual patients with indication of SSI. Our expectations are that the choice and succeeding validation of this proxy variable can provide us with a reliable tool for investigating SSI's in future projects.

The Swedish healthcare registers, especially the Prescribed Drug Register (PDR) and the Patient Register, will make sure that we get an almost complete (>99%) coverage of all relevant SSIs.

The secondary outcomes are: (1) withdrawal or other documented use of any antibiotics for 2 days or more after surgery during the first 30 postoperative days, (2) the number of days with antibiotics during the first 30 days, (3) same as (1) and (2) but up to 90 days after surgery and (d) death during the first 2 postoperative years

These analyses will also be performed with and without adjustment for preoperative antibiotic use 6 months prior to the surgery.

## Sample size

The reoperation rate in Sweden due to infections within the first 2 years after surgery is 1.3%[3] for primary total hip replacements (THRs). This does not include THRs in patients with fracture and other types of surgeries that are more susceptible to infections. We, therefore, assume that the SSI rate in our study population is 2%.[1–3] Similarly, we know from other data that about 0.7% withdraw the antibiotics associated with the primary outcome within a 3-month period prior to surgery. To account for infections unrelated to surgery, we assume that the infection noise rate, that is, non-SSIs, is less than 2%.

### Multicentre power with ultra-clean air

Hospitals with ORs with ultra-clean air may be less susceptible to the effect with an already lower infection rate. Pre-study CFU measurements suggest that approximately 80% of the included ORs are ultra clean. If we assume that the infection rate is 2% or less, and that the effect size is reduced to 25% in an ultra-clean environment, we will need to recruit 22 630 patients in each group, that is, approximately 45 000 patients to attain a power of 80% with a significance level of 0.05. This will be our target population. We expect a very low drop-out rate/missing data in the study, estimated to be <1%. At the interim analysis, the p value will be set 10 times lower at 0.005, that is, if a statistically significant result between the groups is observed at 18 months, the study will be

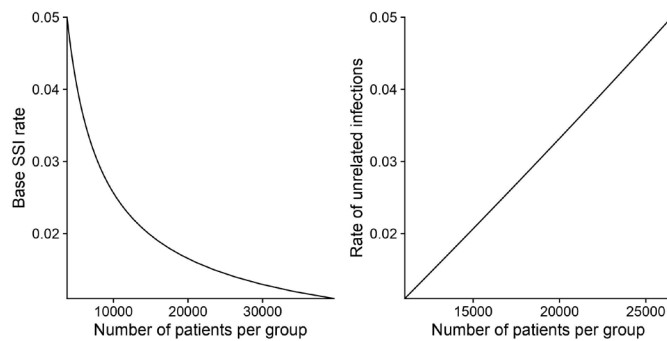

**Figure 3** Graphs showing the impact of base SSI rate and rate of unrelated infections on the required sample size. SSI, surgical site infection.

stopped. See figure 3 and online supplemental appendix 1 for details in R and explanation of the script for the sample size analysis.

## Recruitment

We estimate that with only one centre, such as Danderyd Hospital, recruitment would take >5 years. The hospital operates about 4500 patients each year and as it is unlikely that more than 500 of these will be excluded. We, therefore, anticipate recruiting 4000 patients/year. Performing this study in a multicentre setting is, therefore, crucial. In table 1, the participating centres are presented, and their estimated proportion of patients included.

## Patient and public involvement

Representatives of the Swedish Osteoporosis Society (www.osteoporos.org) and the Swedish Rheumatic Society (www.reumatiker.se) are members of the study steering committee, and participate mainly in discussions regarding plans for reporting and publishing the results of the study.

## METHODS: AN ASSIGNMENT OF INTERVENTIONS
### Allocation

The three PAP devices in each OR will synchronically be randomised in periods of 4 weeks, 6 weeks or 8 weeks to either have the system 'on' or 'off'. The switch will always occur midnight Friday in order to limit the patients exposed to partial effect during the first 2 days after switching status. The system can be programmed to be active (ie, plasma chamber eradicating bacteria) at any given timeframe. The manufacturer of the PAP devices will prepare the randomisation allocation and automatic execution of it. The randomisation sequence is at minimum 8 years long and will be submitted to a third, independent party, responsible for keeping the allocation secret until interim analyses or study end. At the interim analyses, only allocations up to that date will be released.

### Blinding

The on/off only refers to the plasma chamber responsible for the antimicrobial effect. As the machine retains the air flow it will be impossible for staff, surgeons and

patients to determine the status of the machine from the outside. The device will also automatically switch status, where the true status is concealed for all study participants, including other hospital personnel until the end of the study.

## METHODS: DATA COLLECTION, MANAGEMENT AND ANALYSIS
### Data collection methods
For the primary endpoint, the following codes will be used to detect if individual patients have contracted an SSI following surgery. If any of these codes indicate SSI, a chart review will be performed in a second step to verify the outcome:

1. From the Swedish PDR: withdrawal of antibiotics targeting *S. aureus* corresponding to a minimum amount of two defined daily dosages (an estimate provided by the registry for the expected daily dosage). The date of withdrawal will serve as an indicator of treatment start unless inpatient data are available with more granular information. The drug Anatomical Therapeutic Chemical (ATC) codes considered to target relevant bacteria are:
   a. J01CF05 (isoxazolylpenicillin).
   b. J01FF01 (clindamycin).
   c. J04AB02 (rifampicin).
2. From The National Patient Registry: ICD trigger codes indicating postoperative infections:
   a. T793: posttraumatic wound infection, not elsewhere classified
   b. T814: infection following a procedure, not elsewhere classified
   c. T84[5-7]: infection and inflammatory reaction due to internal joint prosthesis, internal fixation device or other internal orthopaedic prosthetic devices, implants and grafts
   d. T874: infection of amputation stump
   e. B9[5,6,8]: bacterial specification
   f. L0[2-4]: cutaneous abscess, furuncle and carbuncle, and cellulitis.
   g. A[24]6: erysipeloid and erysipelas
3. From the National Patient Registry: NOMESCO trigger codes indicating postoperative infections:
   a. Incision abscess: TN[A-H]05
   b. Surgeries due infections: N[A-H]S[0–4,9]9.
   c. Extremities wound revision: Q(CD)B05.
   d. Reoperation for infection: N[A-H]W69.
   e. Vacuum treatment: DQ023.

Both local and national registry data will be used according to availability. For the admission episodes with the code indicators, the admission date is the index date, that is, if an admission occurs after 91 days with a trigger code it will not be considered an indicator of a postoperative infection.

In-hospital information systems will supply information on: (1) patient ID, (2) date(s) of surgery, (3) surgery associated codes, including operated side, (4) OR and (5) in-hospital antibiotics

Both the surgical and medical data records will be retrieved depending on availability. Only centres that can provide the above data will be allowed to participate.

The Swedish National Patient Register includes all in-patient care and outpatient visits in Sweden with discharge codes according to ICD-10, NOMESCO codes and admission/discharge dates.[26]

The Swedish PDR includes any withdrawn prescriptions. Prescriptions that are never withdrawn by patients and drugs bought over the counter without prescriptions are not included. The data fields used were the drug ATC code, number of pills and prescription text.[27]

### Data management
The study data will be securely managed and stored encrypted at a computer within Karolinska Institutet at Danderyd Hospital. No other than the authors stated above will gain access to raw data.

### Statistical methods
The primary outcome is a binary variable where there are three groups. We will use logistic regression where the reference group is the placebo group, and the significance is related to the intervention group. The estimate for the mixed group is only for relating dose effect, that is, the group will not be pooled with either the placebo or the intervention group. Due to the randomisation, we do not intend to have any other covariates as confounders in the model. Similar methodology will be applied to the antibiotic's binary outcome. The number of days with antibiotics will be modelled using a linear regression with the similar interpretation to above regarding predictors. Mortality will be modelled using a Cox proportional hazards model with time since surgery calculated as time to death, migration or 2 years. The Cox model will not contain any covariates due to the randomised study nature. The analysis will be performed by an epidemiologist/statistician in our team (MG) and will be performed as a per-protocol analysis.

We will handle confounding by including only the patient's first surgical procedure. The randomisation process will handle other confounding such as confounding by selection. Procedural confounding will be handled by the external part who has done the randomisation procedure for each PAP device. Regarding dropouts, the healthcare registers and chart review done in the study will ensure a very low (<1%) drop-out rate. Individual patients can also request that they are excluded from the registers and thus from the study, but by experience, this is very rare and will also not affect our ability to analyse our primary and secondary endpoints.

## METHODS: MONITORING
### Data monitoring
At 12 months after study start (17 April 2018), an interim analysis will be performed and the recruitment rate from each centre will be evaluated. The study recruitment

will end once we have reached a minimum of 45 000 patients. During the second half of 2017/early 2018, the data quality of each recruiting centre will be evaluated by extracting data from each centre's hospital information centre.

## EXPECTED RESULTS

Our hypothesis is that the usage of this air purifier significantly lowers the incidence of SSIs after orthopaedic surgery. Since the installation, management and purchase of this kind of machines is nowhere close to the resource intensity of other types of OR ventilation arrangements, it has the possibility of introducing a cost-effective instrument to prevent postoperative infections. Furthermore, this would perhaps benefit especially resource-scarce communities globally.

Secondarily, the large amount of data derived from this study can subsequently be used analysing the effect of other kinds of exposures on the incidence of postoperative infections.

### Ethics and dissemination

The study is being conducted in accordance with the ethical principles of the Declaration of Helsinki, and is approved by the Swedish Ethical Review Authority (2015/1139-31/4).

The interim analysis results from the study will be presented only to the researchers involved unless the study thereafter is interrupted due to significant difference in infection rate between the two groups. Publication in a medical journal will be presented after inclusion of the last patient.

Study participant information will be published on the hospital web site (see online supplemental appendix 2). Due to feasibility reasons in a study with approximately 45 000 study participants, and the very low probability of any adverse effects related to the intervention, no personal consent forms will be collected. However, individual patients can request exclusion from the data analysis.

The EPOS trial is registered at ClinicalTrials.gov.

#### Author affiliations
[1]Karolinska Institutet Institutionen för kliniska vetenskaper Danderyds sjukhus, Danderyd, Sweden
[2]Department of Clinical Sciences, Lund University, Lund, Sweden
[3]Department of Orthopaedics, Hässleholm-Kristianstad Hospitals, Kristianstad, Sweden
[4]Department of Surgical Sciences, Uppsala University, Uppsala, Sweden
[5]Department of Orthopaedics, Uppsala University Hospital, Uppsala, Sweden
[6]Department of Clinical Science Intervention and Technology, Karolinska Institute, Huddinge, Sweden
[7]Department of Orthopaedics and Biotechnology, Karolinska Universitetssjukhuset i Huddinge, Huddinge, Sweden
[8]Department of Surgical and Perioperative Sciences, Umeå University, Umeå, Sweden
[9]Umeå University Hospital, Umeå, Sweden
[10]GHP Ortho Center Stockholm, Löwenströmska Hospital, Upplands Väsby, Sweden
[11]Swedish Hip Arthroplasty Register, Gothenburg, Sweden
[12]Department of Orthopaedics, Institute of Clinical Sciences, Sahlgrenska Academy, University Of Gothenburg, Gothenburg, Sweden

**Contributors** AP operates the trial and led the writing of this manuscript, with contributions from the rest of the authors. TE, MM, HB, NH, SL, TT, IA, SM and P-JK operate the inclusion centres. MG and OS designed the original study and developed the protocol. MG is the responsible statistician and supervises the study. All authors contributed to the editing and redrafting of the manuscript.

**Funding** 8.8 million SEK from the Swedish Research Council has been received for 2017–2020 (grant number: 2017-00198). 1.4 million SEK from ALF (Stockholm County and Karolinska Institute, application number: 20160251, record number: LS 2015–1198) has been received for 2017–2018. We have a discounted price for the rental of the machines but have chosen not to apply for funding from the manufacturers of the plasma air purifier equipment to ensure that the study is independent.

**Competing interests** None declared.

**Patient consent for publication** Not applicable.

**Provenance and peer review** Not commissioned; externally peer reviewed.

#### ORCID iDs
Anders Persson http://orcid.org/0000-0003-1521-4565
Isam Atroshi http://orcid.org/0000-0003-4892-9890

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
