## [Reviewer comments · BMJ Open]

ARTICLE DETAILS

TITLE (PROVISIONAL)	Study protocol: The EPOS-trial - The effect of air-filtration through a plasma chamber on the incidence of surgical site infection in orthopaedic surgery. A randomized, double-blind, placebo-controlled trial
AUTHORS	Persson, Anders; Atroshi, Isam; Tyszkiewicz, Thomas; Hailer, Nils; Lazarinis, Stergios; Eisler, Thomas; Brismar, Harald; Mukka, Sebastian; Kernell, Per-Juan; Mohaddes, Maziar; Sköldenberg, Olof; Gordon, Max

VERSION 1 – REVIEW

REVIEWER	Barchitta, Martina University of Catania, Department of Medical and Surgical Sciences and Advanced Technologies “GF Ingrassia”
REVIEW RETURNED	05-Jan-2021

GENERAL COMMENTS	The manuscript is really very interesting and the study protocol well designed described. I have only a few of suggestion in order to improve the quality of the manuscript and the clarity. Particularly, the time frame of the study should be clarified. If available, results of interim analysis should be discussed. The proxy variable should be briefly described in the aim of the study. The Authors should better justify the choice of this proxy. In the introduction section, other studies conducted to evaluate airborne microbial contamination in OTs during hip and knee replacement surgery, should be mentioned, quoted and used to discuss possible results of the study. I suggest to indicate possible limits and strengths of the study.
---

REVIEWER	Wu, Xiangdong Peking Union Medical College Hospital, Orthopedics
REVIEW RETURNED	27-Jan-2021

GENERAL COMMENTS	Thank you for the opportunity to review this well-designed protocol. This is a protocol for a multicenter, double-blinded, cluster-randomized, placebo-controlled trial to explore whether a new form of air-purifier using plasma-air-purification (PAP) in operating rooms will help to reduce the incidence of SSI after orthopaedic surgery. Although significant progress has been made in preventing and controlling infections, any measure that may help to prevent SSI still should be given high priority. This protocol clearly outlines the proposed study design and analysis methods. However, several concerns exist.
--

	Introduction It would be better if you could provide a simple illustration of how the plasma air-purification (PAP) system works in the Introduction section. Methods 1. Add a CONSORT Flow Diagram to show the phase of this parallel randomized trial. 2. There is a circulation of Intervention group, Control group, and Mixed group. It would be better to add a diagram for this loop. 3. One of my concerns is Data collection, management, and analysis. Actually, many confounding factors may affect the outcome measures and induce bias to the conclusion. Thus, in my opinion, there are a lot of clinical data, including patient-related, surgeon-related, hospital-related, and orthopedic-related, should be collected for analysis. Also, the data management and analysis, as well as subgroup analysis, will be complicated. Do you have any pre-designed subgroup analyses? How will you manage the differences among surgeons and hospitals? I suppose the authors are conducted an RCT under real-world conditions, and the data analysis is critical. 4. Will you consider other second outcomes? Such as re-admission or re-operation due to SSI. Why or why not? I believe this ongoing RCT is well designed and conducted, but the data analysis and interpretation are quite essential. I look forward to the details.
--	---

VERSION 1 – AUTHOR RESPONSE

Reviewer: 1

Dr. Martina Barchitta, University of Catania, University of Catania

At first, thank you, Dr Barchitta for spending your valuable time on reviewing our manuscript. Please find our answers to your comments below. We hope they can clarify your concerns.

Comments to the Author:

The manuscript is really very interesting, and the study protocol well designed described. I have only a few of suggestion in order to improve the quality of the manuscript and the clarity.

Particularly, the time frame of the study should be clarified. If available, results of interim analysis should be discussed.

Thank you, closer specifications of the inclusion time frame have been added to the manuscript. According to the research plan and trial registration, an interim analysis was performed during 2019 including 4400 patients included up until 31st of Dec 2018. These results were presented to the research group in December 2019, but due to concealment issues, there was a consensus in the group to not further spread these results until the end of the study. The results showed, as expected at that time, no statistically significant difference between the intervention groups, and we found no reason to interrupt the study for whatever reason. Furthermore, we have been instructed from BMJ Open not to include any results in the publication of this study protocol.

The proxy variable should be briefly described in the aim of the study. The Authors should better justify the choice of this proxy.

Thank you for this suggestion, a closer description of the proxy variable has been added to the aim section of the manuscript. Please also find an expanded justification of our choice of this proxy variable in the Methods section, Outcomes' paragraph.

In the introduction section, other studies conducted to evaluate airborne microbial contamination in OTs during hip and knee replacement surgery, should be mentioned, quoted and used to discuss possible results of the study.

Thank you for your comment on this. Due to the immensity of this research field, constructing the introduction section of the manuscript has been a challenge, especially regarding to the word limit of this journal. Please find several quoted systematic reviews and meta-analyses in the 4th paragraph of the Introduction section of the manuscript, related to the topic you mention. Please also notice that the orthopaedic surgeries included in our study is not limited to hip and knee replacement surgery, but to all types of orthopaedic surgeries.

I suggest to indicate possible limits and strengths of the study.

Thank you for this suggestion, additions have been made to the five short bullet points in manuscript, also according to recommendations from the Editor of the journal.

Reviewer: 2

Dr. Xiangdong Wu, Peking Union Medical College Hospital

At first, thank you, Dr. Xiangdong for taking your valuable time to review our manuscript. Please find our answers to your comments below. We hope they can clarify your concerns.

Comments to the Author:

Thank you for the opportunity to review this well-designed protocol.

This is a protocol for a multicenter, double-blinded, cluster-randomized, placebo-controlled trial to explore whether a new form of air-purifier using plasma-air-purification (PAP) in operating rooms will help to reduce the incidence of SSI after orthopaedic surgery.

Although significant progress has been made in preventing and controlling infections, any measure that may help to prevent SSI still should be given high priority.

This protocol clearly outlines the proposed study design and analysis methods. However, several concerns exist.

Introduction

It would be better if you could provide a simple illustration of how the plasma air-purification (PAP) system works in the Introduction section.

Thank you for this comment. Please find newly attached rendered figures of the air-purifier in the introduction section.

Methods

1. Add a CONSORT Flow Diagram to show the phase of this parallel randomized trial.

Thank you for this comment. A CONSORT Flow Diagram has now been added among the figures.

Please notice that according to the instructions from the Editor of the journal, no results should be presented, which means that not even up-to-date inclusion numbers can be presented as these are defined as results of the study. If you wish information on these numbers, please contact Dr. Persson as corresponding author.

2. There is a circulation of Intervention group, Control group, and Mixed group. It would be better to add a diagram for this loop.

Thank you for this question. In our opinion, there is no circulation of the groups. Every patient is either randomized to intervention, control or (in rare cases) mixed group, and can only belong to one of

these groups. However, the cluster-design allows for every operating room to change status between air-purifier ON/OFF with varying intervals (4-8 weeks). We chose this design to minimize the risk of allocation bias. Every operating room is considered a cluster, within which exposure status switches according to the randomization program installed in the air-purifiers in that specific room, as described above. We hope that the CONSORT Flow Diagram visualizes this in an adequate way.

3. One of my concerns is Data collection, management, and analysis. Actually, many confounding factors may affect the outcome measures and induce bias to the conclusion. Thus, in my opinion, there are a lot of clinical data, including patient-related, surgeon-related, hospital-related, and orthopedic-related, should be collected for analysis. Also, the data management and analysis, as well as subgroup analysis, will be complicated.

Thank you for this very important notice. We deeply agree with you that many different factors affect the risk of infection after orthopaedic surgery. However, since our study is truly randomized, we assume that both known and unknown confounding factors will be evenly distributed between the groups. Subsequently, as customary in RCT's, these will not be controlled for. Furthermore, also as customary in published RCT's, known confounding factors will be presented in a "Table 1" grouped by intervention status.

Do you have any pre-designed subgroup analyses?

Thank you for this proposal. Yes, subgroup analyses will be performed. In this first paper we will differ between patients operated in ultra-clean (<10 CFU/m³) and regular (>10 CFU/m³) OR's (as described in Methods' section, Intervention paragraph). However, in succeeding analyses we will also sub-group on hospital level, type of surgery and patient characteristics (for ex. age, gender).

How will you manage the differences among surgeons and hospitals? I suppose the authors are conducted an RCT under real-world conditions, and the data analysis is critical.

Thank you for underlining this very important issue. We agree with you that the data management is crucial in this large project. The differences between different surgeons and hospitals (as well as for example seasonal variations in infection incidence) will be handled through the cluster-design of the randomization, as described above. This means that every surgeon's and hospital's patients will be evenly distributed between the intervention groups. The diversity and experience of the research group guarantee a vast experience of conducting RCT's in real-world conditions, as well as management of large amounts of data.

4. Will you consider other second outcomes? Such as re-admission or re-operation due to SSI. Why or why not?

Thank you for also addressing this interesting topic. The EPOS study will result in a large amount of data, why several secondary analyses and projects are in a planning stage. Re-operations due to SSI are already included in this first project, as they are indicated by the NOMESCO codes (please see Methods' section, Data collection methods' paragraph). Re-admissions will to a large extent be included, since they often result in an ICD or NOMESCO code indicating SSI, but re-admission as itself is not included in this first analysis. However, it would be interesting to include in an upcoming project regarding cost-effectiveness of these air-purifiers.

I believe this ongoing RCT is well designed and conducted, but the data analysis and interpretation are quite essential. I look forward to the details.

VERSION 2 – REVIEW

REVIEWER	Barchitta, Martina University of Catania, Department of Medical and Surgical Sciences and Advanced Technologies “GF Ingrassia”
REVIEW RETURNED	20-Jun-2021

GENERAL COMMENTS	The manuscript is interesting. The article reports the protocol of a multicentre, double-blinded, cluster-randomized, placebo-controlled trial in order to assess the efficacy of a PAP devices that clean the air in operating rooms on SSI's rate. The methodology is appropriate. I have some concerns about the primary outcome. Particularly, it is not clear if only S. aureus infections will be monitored. Furthermore, I think that a possible limit of the assessment is related to postoperative infections that will not be managed in hospitals participating in the trial and thus a specific surveillance post-intervention must be implemented. Please check and revise period of data monitoring. Finally, I suggest to include a section with expected results and a section with strengths, authors should further highlight the importance of expect results, and limits, in particular I think that a specific surveillance protocol could be implemented in order to collect other relevant risk factors that could be influence SSI's rate, as patients factors and those related to hospital, procedures and other.
--

REVIEWER	Wu, Xiangdong Peking Union Medical College Hospital, Orthopedics
REVIEW RETURNED	26-Nov-2021

GENERAL COMMENTS	My concerns have been cleared
-------------------------------

VERSION 2 – AUTHOR RESPONSE

Reviewer: 1

Dr. Martina Barchitta, University of Catania, University of Catania

At first, thank you once again, Dr Barchitta for spending your valuable time on reviewing our manuscript. Please find our answers to your comments below. We hope they can clarify your concerns

Comments to the Author:

The manuscript is interesting. The article reports the protocol of a multicentre, double-blinded, cluster-randomized, placebo-controlled trial in order to assess the efficacy of a PAP devices that clean the air in operating rooms on SSI's rate. The methodology is appropriate. I have some concerns about the primary outcome. Particularly, it is not clear if only *S. aureus* infections will be monitored.

Thank you for this comment. Surgical site infections caused by all kinds of bacteria will be regarded as postoperative infections. The reason for only monitoring *S.aureus*-specific antibiotics is that other types of antibiotics (for example ciprofloxacin and amoxicillin) are often used in Sweden against other types of common infections (i.e. urinary tract infection and airway infection), and thus would implement considerable amount of misclassification bias, and thereby limit study power. Please notice that the primary outcome is a composite made out of three parts where prescription of antibiotics is only one part. ICD- and NOMESCO-codes specifying diagnoses and surgical procedures will also be

used, and non-S.aureus-infections will be identified this way. Please find the specification regarding this in the Methods' section, Data collection methods' paragraph. We expect that many non-S.aureus-infections will be identified through antibiotic prescriptions since both clindamycin and rifampicin are used against C. acnes and other types of biofilm-producing bacteria in Sweden.

Furthermore, I think that a possible limit of the assessment is related to postoperative infections that will not be managed in hospitals participating in the trial and thus a specific surveillance post-intervention must be implemented. Please check and revise period of data monitoring.

Thank you for this comment, as it is an important one. It is correct that some of the postoperative infections will not be managed at the participating hospitals. Some of them will not even be managed at any hospital at all, but rather at a primary health care center or polyclinic. They will still be monitored in the study, since all antibiotic prescriptions, ICD- and NOMESCO-codes from all health care givers in the country are reported to The Swedish Prescribed Drug Register and The Swedish National Patient Register, from where we will collect the data. Thus, none of them will be missed, as the completeness and coverage of these codes in Sweden is almost perfect, i.e close to 100%. Please find the specification regarding this in the Methods' section, Data collection methods' paragraph.

Finally, I suggest to include a section with expected results and a section with strengths, authors should further highlight the importance of expect results, and limits, in particular I think that a specific surveillance protocol could be implemented in order to collect other relevant risk factors that could be influence SSI's rate, as patients factors and those related to hospital, procedures and other.

Thank you for this final suggestion. Please find an added section in the manuscript, with expected results and their importance for the orthopaedic society. As earlier requested by the Journal Editor, please also find strengths and limits in the Strengths and limitations of this study's section, as a bullet point list. Thank you also for the important comment on other relevant risk factors. We deeply agree with you that many different factors affect the risk of infection after orthopaedic surgery. However, since our study is randomized, we assume that both known and unknown confounding factors will be evenly distributed between the groups. Subsequently, as customary in RCT's, these will not be controlled for. The differences between different surgeons and hospitals (as well as for example seasonal variations in infection incidence) will be handled through the cluster-design of the randomization. This means that every surgeon's and hospital's patients will be evenly distributed between the intervention groups. Furthermore, also as customary in published RCT's, known confounding factors will be presented in a "Table 1" grouped by intervention status.

Reviewer: 2

Dr. Xiangdong Wu, Peking Union Medical College Hospital

Comments to the Author:

My concerns have been cleared.

Thank you, Dr Xiangdong, for your earlier comments. We are pleased to read that your concerns have been cleared.

VERSION 3 – REVIEW

REVIEWER	Barchitta, Martina University of Catania, Department of Medical and Surgical Sciences and Advanced Technologies “GF Ingrassia”
REVIEW RETURNED	03-Jan-2022
GENERAL COMMENTS	Thank you for providing the new version of the manuscript taking into account all my comments and suggestions